

# European climate change at global mean temperature increases of 1.5 and 2°C above pre-industrial conditions as simulated by the EURO-CORDEX regional climate models

Erik Kjellström[1, 2], Grigory Nikulin[1], Gustav Strandberg[1], Ole Bøssing Christensen[3], Daniela Jacob[4], Klaus Keuler[5], Geert Lenderink[6], Erik van Meijgaard[6], Christoph Schär[7], Samuel Somot[8], Silje Lund Sørland[7], Claas Teichmann[4] and Robert Vautard[9]

[1]Rossby Centre, Swedish Meteorological and Hydrological Institute (SMHI), 601 76 Norrköping, Sweden
[2]Department of Meteorology (MISU), Stockholm University, 106 91 Stockholm, Sweden
[3]Danish Danish Climate Centre, Danish Meteorological Institute (DMI), Copenhagen, Denmark
[4]Climate Service Center Germany (GERICS), Helmholtz-Zentrum Geesthacht
[5]Environmental Meteorology, Brandenburg University of Technology, Cottbus, Germany
[6]Royal Netherlands Meteorological Institute (KNMI), De Bilt, The Netherlands
[7]Institute for Atmospheric and Climate Science, ETH Zürich. Universitätstrasse 16, 8092 Zürich, Switzerland.
[8] CNRM UMR 3589, Météo-France/CNRS, Toulouse, France
[9]Laboratoire des Sciences du Climat et de l'Environnement, IPSL, CEA/CNRS//UVSQ, Gif sur Yvette, France

*Correspondence to*: Erik Kjellström (erik.kjellstrom@smhi.se)



**Abstract.** We investigate European regional climate change for time periods when the global mean temperature has increased by respectively 1.5°C and 2°C compared to preindustrial conditions. Results are based on regional downscaling of transient climate change simulations for the 21st century with global climate models (GCMs) from the fifth phase Coupled Model Intercomparison Project (CMIP5). We use an ensemble of EURO-CORDEX high-resolution regional climate model (RCM) simulations undertaken at a computational grid of 12.5 km horizontal resolution covering Europe. The ensemble consists of a range of RCMs that have been used for downscaling different GCMs under different forcing scenarios. The results indicate considerable near-surface warming already at the lower 1.5°C warming. Regional warming exceeds that of the global mean in most parts of Europe, strongest in northernmost parts of Europe in winter and in southernmost parts of Europe together with parts of Scandinavia in summer. Changes in precipitation, that are less robust than the ones in temperature, include increases in the north and decreases in the south with a borderline that migrates from a northerly position in summer to a southerly one in winter. Some of these changes are seen already at 1.5°C warming but larger and more robust at 2°C. Changes in near-surface wind speed are associated with a large spread between individual ensemble members at both warming levels. Relatively large areas over the North Atlantic and some parts of the continent shows decreasing wind speed while some ocean areas in the far north show increasing wind speed. The changes in temperature, precipitation and wind speed are shown to be modified by changes in mean sea level pressure indicating a strong relationship with the large-scale circulation and its internal variability on decade-long timescales. By comparing to a larger ensemble of CMIP5 GCMs we find that the RCMs can alter the results leading either to attenuation of amplification of the climate change signal in the underlying GCMs. We find that the RCMs tend to produce less warming and more precipitation (or less drying) in many areas in both winter and summer.



# 1 Introduction

A main aim of the Paris agreement within the UNFCCC (United Nations Framework Convention on Climate Change) is to keep the increase in the global average temperature well below 2°C above pre-industrial levels and to pursue efforts to limit the temperature increase to 1.5°C above pre-industrial levels (UNFCCC, 2015). While the agreement comes into power 2020 we observe ongoing global warming with the most recent years continuing the long-term warming trend of the last decades (WMO, 2016). Regional and local impacts of global warming are already seen and there is a strong concern that these impacts will become worse with stronger future climate change (IPCC, 2014). However, exactly how strong these impacts will be at different warming levels is uncertain as information about the climate change signal on a regional level is scarce. Despite some efforts that have been made looking at possible climate change at 1.5°C or 2°C global warming and on comparing differences at these global warming levels (e.g. Vautard et al., 2014; Fischer and Knutti, 2015; Schleussner et al., 2016; King and Karoly, 2017) detailed information about regional climate change is largely missing for scenarios reflecting 1.5°C global warming (e.g. Mitchell et al., 2016)..

Much of the available information about future regional climate change comes from global climate models (GCMs). The most comprehensive set of GCM data is that of the CMIP5 (5th phase of the Coupled Model Intercomparison Project, e.g. Taylor et al., 2012) consisting of more than 30 GCMs. An advantage with GCMs is that they can provide regional information for all areas in the world. A limitation, however, is the fact that they are commonly operated at relatively coarse horizontal resolution (most often at 100-200 km grid spacing). This implies that land-sea contrasts and land surface properties including mountain height are only described in a coarse way and that important phenomenon like mid-latitude cyclones and mesoscale processes are handled in a rudimentary way. Dynamical downscaling with regional climate models (RCMs) is one way of providing high-resolution climate information that better account for regional to local scales and thereby add value compared to the GCM (e.g. Rummukainen, 2010; Sørland et al., 2017). For Europe relatively large data sets of RCM scenarios have previously been put forward within the context of European research projects including PRUDENCE (Christensen et al., 2007; Déqué et al., 2007) and ENSEMBLES (van der Linden et al., 2009; Déqué et al., 2012; Kjellström et al., 2013). In recent years RCMs have been operated in the framework of CORDEX (Coordinated Regional climate Downscaling EXperiment, e.g. Jones et al., 2011; Gutowski et al., 2016). For Europe, in particular, this means that an unprecedented data set of RCM scenarios at 50 and 12.5 km horizontal resolution is available from the EURO-CORDEX project (Jacob et al., 2014). Previous work have shown that the high-resolution 12.5 km simulations add value compared to the 50 km simulations, in particular in terms of representing extremes like heavy-precipitation events (e.g. Kotlarski et al., 2014; Prein et al., 2015). Other studies describing evaluation of different important near-surface variables in the EURO-CORDEX RCMs in the recent past climate include those of Smiatek et al. (2016); Knist et al. (2016) and Frei et al. (2017).



The relatively large ensemble of EURO-CORDEX high-resolution RCM climate change scenarios constitutes a valuable data set for impact studies. Some of these simulations and from the earlier ENSEMBLES project have been used for considerations of climate change at different warming levels (e.g. Vautard et al. 2014; Maule et al. 2017) and in impact studies (e.g. Alfieri et al. 2015; Donnelly et al. 2017). However, previous studies have either been based on earlier RCM ensembles or only on smaller subsets of the full EURO-CORDEX set of RCM simulations. In this study we therefore focus on how the European climate may change at the 1.5°C or 2°C global warming levels in the larger set of EURO-CORDEX simulations at 12.5 km grid spacing. Specifically, we address at which of the two warming levels we can detect significant climate change compared to a reference period in the end of the 20th century and to what extent changes at the two warming levels differ which is important for mitigation considerations. We also show how different sources of uncertainty influence the climate change signal and discuss how the EURO-CORDEX simulations relate to the larger CMIP5 GCM ensemble.

## 2 Methods and material

### 2.1 Climate model simulations

We use RCM data from eighteen EURO-CORDEX simulations for the European area; see Table 1 and Fig. 1. All simulations have been done with forcing following RCP8.5 (Representative Concentration Pathway, see Moss et al., 2010). The chosen simulations allow us to address the impact of different driving GCM on the resulting climate change signal. In addition, the impact of choice of different RCMs can be investigated both for the three-member RCM ensembles downscaling MPI-ESM-LR-r1, EC-Earth-r12, HadGEM2-ES and CNRM-CM5 and for the two-member ensemble downscaling IPSL-CM5A-MR. Furthermore, as three members of EC-EARTH and two members of MPI-ESM-LR are included, also the role of internal natural variability can be addressed. The simulations have been chosen based on the availability of data at the Earth System Grid Federation (ESGF) facility.

RCM results are set in a larger context by comparing to 31 simulations from the CMIP5 multimodel GCM ensemble (Table 2). In addition to the GCM simulations listed in Table 1, also the first ensemble members of the other CMIP5 GCMs are assessed for seasonal mean changes in precipitation and temperature. In this way we can investigate how the smaller subset of GCMs that provides input for the RCMs replicates the larger CMIP5 GCM ensemble. We can also look at if, and to what extent, the RCMs change the climate change signal of the underlying GCMs. Comparisons are done for a number of regions in Europe previously used in a large number of studies (e.g. Rockel and Woth, 2007), see Fig. 1.



## 2.2 Calculation of warming levels

We investigate periods for which the global mean near surface temperature is 1.5 and 2.0˚C above preindustrial conditions (hereafter referred to as SWL1.5 and SWL2). As the temperature in true pre-industrial, i.e. pre-1750, conditions are not known (cf. Hawkins et al., 2017; Schurer et al., 2017), we use the simulated climate from the GCMs for 1861-1890 as a proxy. For each GCM we then identify the first period when the 30-year running mean global temperature reaches 1.5 respectively 2.0˚C above that of the pre-industrial period. These thirty-year time slices (see Table 2) are used for the analyses in the study. For comparing future climate change we then use the period 1971-2000 as our reference in the regional climate model simulations. From observations we note that the global warming between 1861-1890 (pre-industrial) and 1971-2000 (reference) is 0.41K according to HadCRUT4 (Morice et al., 2012) implying that future temperature changes above 1.1 and 1.6˚C represents a regional warming exceeding the global average for the two warming levels.

## 3 Results

Here we focus on comparing simulated changes at SWL1.5 and SWL2 for seasonal mean near-surface temperatures, precipitation and wind speed over Europe for winter (December-February, DJF) and summer (June-August, JJA). Climate change in the EURO-CORDEX ensemble is presented showing both ensemble means and information about robustness of the change as indicated when at least 14 out of the 18 ensemble members (i.e. more than 75%) agree on the sign of change in a variable.. First, however, we show how changes in mean sea level pressure (MSLP) differ between the individual ensemble members as these changes will be shown to have strong impacts on changes in the other variables.

### 3.1 Simulated changes in MSLP

Figs. 2 and 3 show the changes in MSLP in each ensemble member at SWL2 for winter and summer respectively. It stands clear that there are considerable differences between the different simulations and that these differences are closely connected both to the choice of GCMs (e.g. comparing CNRM-CM5-driven simulations with those driven by HadGEM2-ES) but also to the choice of ensemble member (as illustrated by the two realizations of MPI-ESM-LR or the three EC-EARTH members). Further, we note that there are some differences also due to the RCM. This can be seen from the six panels showing the RACMO, CCLM and RCA4 simulations downscaling EC-EARTH-r12 and HadGEM2-ES. The most pronounced difference in winter is the stronger increase in MSLP in southern and central Europe in the HadGEM2-ES-driven CCLM simulation compared to the two others (Fig. 2). Also for summer this CCLM simulation differs compared to the two other RCMs in showing an increase in MSLP in large parts of Eastern Europe and the Baltic Sea region (Fig. 3).

In winter we note that the strong north-south pressure gradient over the North Atlantic is changed differently in the different simulations (Fig. 2). In the southern half of the domain in the MPI-ESM-LR-r1 driven simulations there is a weakening in



this pressure gradient while it is intensified in the north. This indicates a northward shift in the storm track with less (more) mild air being advected in over central and southern Europe (northern Scandinavia) from the Atlantic. Similar patterns are seen in the simulations driven by HadGEM2-ES, NorESM1-M and in EC-EARTH-r1. Contrastingly, EC-EARTH-r12 shows a completely different pattern with a strengthening of the north-south pressure gradient, albeit with no major relocation of it,

indicating a strengthening of the westerlies over the North Atlantic. Also CNRM-CM5 indicates a strengthening of the gradient although not as strong. The MPI-ESM-LR-r2-driven simulation and the EC-EARTH-r3-driven run both show decreasing MSLP over the British Isles and in a band in over the European continent indicating a southward shift of the storm track. Finally, IPSL-CM5A-MR shows a very different pattern with lower pressure in general over large parts of northern Europe indicating a stronger low pressure activity in this area.

Also for summer the change patterns differ. Several simulations indicate a strengthening and/or northward displacement of the subtropical high (both MPI-ESM-LR members, all EC-EARTH members and NorESM1-M). In MPI-ESM-LR-r1 the strengthened subtropical high is also associated with a decrease in pressure in the northernmost part of the Atlantic and over Scandinavia. This pattern is indicative of a northward shift of the storm track in summer. Five out of the six GCM

simulations with a strengthening of the subtropical high show a reinforcement of this signal with warming as the MSLP anomalies are larger at SWL2 than at SWL1.5 (not shown). A similar pattern with reinforcements in in MSLP changes when looking at SWL2 compared to SWL1.5 is not generally seen in winter. This contrast between the two seasons indicates that changes in winter are more associated with internal variability while summertime changes are to a larger degree associated with long-term global warming.

**3.2 Simulated changes in near-surface temperature**

Warming is manifested in all seasons as exemplified for winter and summer in Figs. 4 and 5. A comparison of the climate change signal at the two warming levels shows considerably larger changes at SWL2 than at SWL1.5. A number of regional features stand clear from the figures. This includes a stronger warming in winter than in summer in large parts of northeastern Europe. In summer the strongest warming is found in the south and southeast but also parts of Scandinavia

show strong warming. This is consistent with findings of Vautard et al. (2014) who analysed a different set of simulations and scenario. Changes are generally smaller over the oceans than over land areas with the exception of some parts of the northern seas that show very strong warming mainly in winter but also to some extent in summer. This strong warming over the northern seas can to a large degree be attributed to reduction in sea ice in the warmer climate. The stronger warming in summer over the Baltic Sea than over its surroundings cannot be directly related to changes in sea-ice as there is none in the

Baltic Sea in summer. We have not investigated the reason for the Baltic Sea warming in detail here but we note that it is larger in some GCM-driven experiments than others (not shown) so it is likely that the boundary forcing from the GCMs is the cause.



Figures 4 and 5 reveal that temperature increase is a highly robust feature of the RCM-GCM combinations assessed here as basically all eighteen simulations indicate increasing temperatures in both seasons already at SWL1.5. It is only in winter that a few (1-3) ensemble members display a weak decrease at SWL1.5 over parts of Eastern Europe and Scandinavia while almost all individual simulations show warming in all of these areas too at SWL2 (not shown). Apart from these exceptions

over the continent a few simulations also show the absence of warming over parts of the Atlantic west of the British Isles as a result of a weaker warming in the underlying GCMs in this area). Despite the agreement on sign there are still large differences between individual simulations in some areas (not shown). This is most notable in the far north over ocean areas in winter which is the area in Europe warming the most (cf. Fig. 4). Apart from the far north we also note relatively large spread in southeastern Europe in winter at both warming levels. A closer look at the individual simulations reveals that the

three MPI-ESM-LR-r1-driven simulations all give very modest warming, or even local cooling, both for SWL1.5 and SWL2 (not shown). Recalling the changes in MSLP (Fig. 2) we interpret this as a consequence of the changing large-scale circulation with weaker southwesterlies bringing less mild Atlantic air in over Europe. Similarly, we can interpret the larger temperature increase in the EC-EARTH-r12-driven RACMO simulation over large parts of Europe compared to the corresponding EC-EARTH-r1-driven one with the changes in MSLP discussed above. Apart from these GCM-driven

differences we also note differences arising from choice of RCMs. For instance we note that RCA4 shows stronger warming than CLM in Eastern Europe in winter when forced by EC-EARTH-r12 as does RACMO in the HadGEM2-ESM-driven one (not shown). Similarly, ALADIN shows stronger warming in summer in south-eastern Europe than both CCLM and RCA4 when forced by CNRM-CM5. These differences between the RCMs indicate some systematic difference between them and how they respond to changes in the large-scale forcing.

**3.3 Simulated changes in precipitation**

Precipitation changes in the analysed simulations follow the well-known pattern for Europe with tendencies for increasing precipitation in the north and decreases in the south on an annual mean basis (not shown). The borderline between increasing and decreasing precipitation migrates from a southerly position in winter (Fig. 4) to a northerly position in summer (Fig. 5). Changes generally increase over time and the extent of areas showing robust changes increases from SWL1.5 and SWL2.

However, in some areas changes at SWL2 are smaller than those at SWL1.5. As an example the Iberian Peninsula and the adjacent North Atlantic show strong increases in wintertime precipitation already at SWL1.5 while there is no additional increase (or even a weaker signal indicating decrease between the two periods) at SWL2. Compared to the findings for temperature, precipitation changes are less robust. Notably, there are relatively large areas without hatching on the maps where different RCM simulations show either an increase or a decrease in precipitation. We also note that in areas where

there is partial consensus of 14-15 models or more on sign of the change there can still be large uncertainties related to the amplitude of the change (Fig. 6).

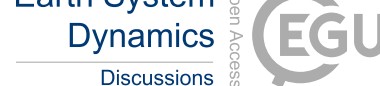

In some more detail it is clear that some of the differences in precipitation response are strongly related to changes in the large-scale circulation. As an example, a comparison of Fig. 6 and Fig. 2 reveals that decreasing precipitation south of Iceland and along parts of the Norwegian coast in the HadGEM2-ES-driven simulations can be connected to the weaker north-south pressure gradient in this area. Contrastingly, the stronger N-S pressure gradient over the Atlantic in the EC-EARTH-r12-driven simulations, and to a lesser extent in the CNRM-CM5-driven simulations, leads to substantial increases in precipitation over the British Isles and southern Norway. The northward shift in the storm track in summer (cf. Fig. 3) is reflected by strong increases in precipitation in parts of Scandinavia (Fig. 5). In southern and central Europe, on the other hand, there is a reduction in precipitation in connection with the northward displacement of the subtropical high and increasing MSLP over Europe. Again, there are large differences between individual RCMs also when forced by the same driving GCM simulation. Examples include stronger increases in precipitation in RCA4 compared to CCLM and RACMO in northern Scandinavia (not shown).

## 3.4 Simulated changes in near-surface wind speed

The simulated climate change signal in mean near-surface wind speed is generally not robust over Europe. Decreases are seen over parts of the North Atlantic and the Mediterranean in winter (Fig. 4) and over parts of the North Atlantic and western Europe in summer (Fig. 5), confirming the results of Tobin et al. (2016). Increases are seen over some northern ocean areas. These are strongest in winter but can to some extent be seen in all seasons. A closer look at the individual simulations reveals that there are strong connections to the variability in the large-scale circulation as indicated by changes in the MSLP pattern. Notably, the weakening and northward shift in the N-S pressure gradient in the Had-GEM2-ES-driven simulations is reflected in a considerable decrease in wind speed in large parts of the area while the sharpening of the gradient in the EC-EARTH-r12-driven simulations lead to strong increases in wind speed in the area of the British Isles (compare Fig. 2 and Fig. 7). Apart from these changes that are related to changes in the large-scale circulation, there are also other wind speed changes. Figure 7 reveals that the local increases over parts of the northern oceans are seen in most simulations although at slightly different locations. There are no common changes in MSLP that can explain this pattern. On the other hand, we note strong increases in near-surface temperature in these areas in the models (not shown). This strong relation between near-surface temperature and winds indicates that changing surface conditions are important here. Likely, the reduction of sea-ice and the associated higher temperatures lead to a less stably stratified planetary boundary layer that thereby becomes more favourable for downward mixing of momentum leading to higher wind speed close to the surface. Also in summer, changes in sea-ice and associated changes in sea surface temperatures may contribute to increasing wind speed over the Arctic Ocean areas in some simulations (not shown). However, we also note similar differences in some simulations over the Baltic Sea where sea ice cannot be the reason for summertime differences. Changes in wind speed are more pronounced in some areas at SWL2 than at SWL1.5 indicating that we are looking at a manifestation of long-term climate change. However, we note that the areas where models agree upon sign of change in wind speed do not get considerably larger at SWL2 and that large areas don't show any systematic changes in wind speed reflecting the importance



of internal variability. In summary, the results indicate that it is highly uncertain what may happen to wind speed in this region when global warming continues.

## 4 Discussion

### 4.1 Is there a detectable climate change signal in the EURO-CORDEX ensemble at 1.5 and 2°C global warming?

The results of the 18 RCM simulations analysed here show increasing temperatures, changing precipitation patterns and also some changes in seasonal mean wind speed. These changes are more or less robust for the different variables. Temperature increases are simulated in all parts of Europe for all seasons by almost all models already at SWL1.5. Precipitation changes show larger model spread; noteworthy are uncertainties even in sign of change in the seasonally migrating area between increasing precipitation in the north and decreasing precipitation in the south. For wind speed there are also large

uncertainties with different models showing very different response patterns. We note that for the studied variables the ensemble mean changes at SWL2 are generally larger than those at SWL1.5. This is always the case for temperature while for precipitation and wind speed there are local exceptions to this. Differences between individual ensemble members are often large, sometimes larger than the overall climate change signal at SWL1.5 and SWL2. It is evident that a clear robust climate change signal has not emerged in all variables, seasons and regions studied here. This finding is in accordance with

earlier studies that have also shown different times of emergence of a regional climate change signal (e.g. Giorgi and Bi, 2009; Hawkins and Sutton, 2012; Kjellström et al., 2013).

The results show that the large-scale circulation has an important role in determining the actual climate change signal in any individual simulation. For instance, it stands clear that stronger westerlies in some simulations leads to milder and wetter

conditions over parts of the continent while weaker westerlies lead to less precipitation along the western coastlines. This is in concert with previous studies showing a similar dependence (e.g. Van Ulden and van Oldenborgh, 2006; Kjellström et al., 2011; Kjellström et al., 2013). Differences in the large-scale circulation over decade-long climate simulations are not necessarily a sign of climate change but rather a manifestation of the large internal variability of the climate system that can be pronounced on a regional scale (e.g. Hawkins and Sutton, 2009). As our results are based on a relatively small number of

GCM simulations we are limited in to what degree the ensemble captures the full uncertainty. Larger ensembles consisting of multiple simulations with one, or preferably many models, would give better opportunity to sample this uncertainty (e.g. Deser et al., 2012; Aalbers et al., 2017). It is clear that the natural variability with its impacts on the large-scale circulation is a major cause of uncertainty. This is highly pronounced when it comes to assessing climate change signals at any of the two warming levels discussed here as changes are, even if robust and seen in most simulations, still not necessarily exceeding the

natural variability.



**4.2 Timing for reaching 1.5 and 2˚C above pre-industrial conditions**

An alternative approach to the one used here for investigating climate change at the time of 1.5 and 2˚C warming would be to use scenarios in which the climate system reaches a new equilibrium at the requested warming levels. This could for instance be closer to the end of the century in scenarios with rapidly decreasing forcing and eventual stabilization of the

climate. A difficulty with that approach is that different GCMs with different climate sensitivities may either, not reach the warming levels or, exceed them. The definition of warming levels used here assures that the global mean warming for the investigated periods is exactly 1.5 and 2˚C above pre-industrial conditions as simulated by the models. The choice of extracting this information from a transient simulation implies that there will be trends in the time slices that may influence the results (Bärring and Strandberg, 2017). For instance, interannual variability may be artificially augmented in case of a

long-term increasing (or decreasing) trend. Such trends could be removed before investigating interannual variability or extreme conditions that may be sensitive to increased temporal variability. However, for this study we have chosen not to do this as we focus on long-term seasonal averages. Another potential problem with the transient approach is when results are going to be used in impact studies for which there may be other important time constraints. A certain level of global climate change may have very different regional signatures at different timings. For instance, Maule et al. (2017) shows that if the

time it takes until a certain warming level is reached is longer (as a result from less strong forcing in RCP4.5), the regional climate change signal in Europe is less strong than if the level is reached quickly (as result from strong forcing in RCP8.5). Apart from such differences in regional climate response, impacts will be different if changes are quick or slow depending on the resilience of the considered society or ecosystem.

Here, we present information about when the two specific warming levels are reached given the data used in the study. A benefit of this transient, non-stabilized approach is that it represents conditions that may be more representative for what happens if we do not meet the 2˚C target (or 1.5˚C for that matter). Even if global warming will be more than 2˚C it may be valuable to look at SWLs in an adaptation context, as a level of climate change that we will have to adapt to on our way to the even warmer climate beyond 2˚C. In that case this approach is a way to shift the perspective from the relatively uncertain

level of climate change at a specific point in time, to a more certain level of climate change at an uncertain point in time.

Partly due to their different climate sensitivity the CMIP5 GCMs reach the different warming levels at different points in time. For the 31 RCP8.5 runs in Table 1 the central years of the 30-year periods range between 2009 and 2043. The subset of eight GCMs that has been downscaled in EURO-CORDEX and further assessed here shows central years ranging between

2016 and 2029. Therefore, it is clear that the chosen subset doesn't sample the full range of climate sensitivities in the GCMs.





Some GCMs have been run several times to sample the natural variability of the system and usually these ensemble members show slightly different results. The largest ensemble of one GCM in the CMIP5 data set is the CSIRO model with 10 different members (member number 1 is shown in Table 1). The central year for reaching the 1.5˚C warming level in that ten-member ensemble ranges between 2027 and 2035. For the corresponding 2˚C level it ranges between 2041 and 2046.

These relatively smaller intervals, compared to those of the CMIP5 multi-model ensemble discussed above, indicate that the simulated natural variability of the global mean temperature is a smaller source of uncertainty than that of the climate sensitivities as represented by the different GCMs. This does not, however, imply that natural variability on the regional scale is not important as a source of uncertainty (as discussed in Ch. 4.1).

In addition to climate model sensitivity and natural variability also different forcing plays a role for when a certain warming level is reached. We note that the 30-year time slices used in the analysis here partly overlaps between the two time windows. For the RCP8.5 scenarios central years between the two periods differ by between 18 and 10 years in any of the GCM simulations indicating that at least 12 years are common for the two time slices for any given simulation while for some model simulation even up to 20 years are the same. Clearly, the two samples are more similar compared to if they were

taken as time slices more separated from each other. This similarity has implications for how to assess differences between the periods in a statistically rigorous way as data in the two samples are not independent.

All 31 GCMs in Table 1 have also been run for RCP4.5. For that scenario SWL1.5 is reached between 2008 and 2061 in the different models (not shown). SWL2, on the other hand, is reached at 2024 by the first model while for five of the models it

is not reached at all during the 21$^{st}$ century. For the 26 simulations that do SWL2 under RCP4.5 the timing for any one of them differ from the time when the same simulation reaches SWL1.5 by between 35 and 14 years indicating that in some cases there is no overlap between the two warming levels but in some cases up to 16 years are common. Clearly, there is an impact on the similarity of the results between the two time slices depending on which scenario that is used.

### 4.3 How representative are the results from the EURO-CORDEX ensemble?

In this section we discuss how the above-mentioned RCM-based results relate to the underlying GCMs and to the larger CMIP5-ensemble by showing scatter plots for changes in temperature and precipitation. We present scatter plots for Scandinavia and Eastern Europe as these are the two areas in Fig. 1 that shows the strongest changes in temperature: in winter in Scandinavia and in summer in Eastern Europe. Precipitation shows an increase in Scandinavia in both winter and summer. In Eastern Europe it increases in winter while different models show either increases or decreases in summer.

Comparing and contrasting these areas give a good picture of changes in some of the climate regimes of Europe. In Table 3 we present summary statistics for SWL1.5 in the subregions defined in Fig. 1.





Figure 8 and Table 3 shows that simulated temperature changes in Scandinavia are larger in winter (1.7˚C) than in summer (1.5˚C). For comparison with preindustrial conditions we remind ourselves that this change is to be added to the 0.41˚C increase in global mean temperature between 1861-1890 and 1971-2000. For the Scandinavian region past changes are larger, data representing Sweden shows that warming over this period is almost 1˚C (data taken from www.smhi.se)

indicating a warming of more than 2.5˚C compared to pre-industrial conditions already at SWL1.5. Figure 8 shows that the simulated future warming is stronger in Scandinavia compared to the global mean warming already at SWL1.5 and even more pronounced at SWL2 for a majority of the simulations. For precipitation the majority of the simulations indicate that it will become wetter in both winter and summer which is seen in many simulations already in SWL1.5 and more clearly in SWL2. However, for both seasons there are also simulations showing only little change or even decreasing precipitation.

It is clear that the spread between the simulations becomes larger at SWL2 compared to SWL1.5 both in temperature and precipitation based on the full CMIP5 model ensemble. We note that the RCM-simulated changes in temperature and precipitation mostly lies within the range of those as simulated by the underlying GCMs and by the larger CMIP5 ensemble. However, it is also clear that the range spanned by the RCM ensemble (or that spanned by the underlying GCMs) is more

limited compared to the full CMIP5 ensemble. Comparing individual simulations reveals that the RCMs do modifies the climate change signal from the underlying GCM. There are, however, large differences in how large these modifications are. For instance, the REMO RCM only changes the climate change signal from the MPI-ESM-LR model marginally in all four cases while all three RCMs that have downscaled HadGEM2-ES change the results significantly in summer. In the latter case it is even the question of changing sign in the precipitation signal; from a decrease in HadGEM2-ES to an increase in

the RCMs. A similar discrepancy between HadGEM2-ES and RCA4 was found also in a RCA4 simulation at 50 km horizontal resolution by Kjellström et al. (2016). They also found wetter conditions in RCA4 compared to a range of other GCMs it has downscaled indicating that the hydrological cycle is more sensitive to the increasing temperatures in this regional climate model. It is also noted that HadGEM2-ES has a very strong increase in sea surface temperatures (SSTs) over the Baltic Sea, as indicated by the local maxima in near-surface warming (not shown). Large SST changes in this region

have previously been shown to have a very strong impact on regional climate modelling results (e.g. Kjellström and Ruosteenoja, 2007). As coarse-scale GCMs have a fairly poor representation of the Baltic Sea care should be taken when analyzing results from these models and preferably a coupled regional climate model system should be used (Kjellström et al., 2005). Apparently, many of the RCM simulations assessed here show larger precipitation increases (or smaller decreases) compared to the underlying GCMs for the Scandinavian domain as also indicated by the ensemble mean statistics.

For Eastern Europe Fig. 9 shows that simulated changes in temperature are slightly larger in summer than in winter at both SWLs. Also the spread is larger in summer as a number of models give very strong temperature increases (among these are HadGEM2-ESM that has been downscaled by the RCMs). For precipitation the simulations reveal an uncertainty not just in amplitude but also in sign of change in both winter and summer with models indicating either increase or decrease. The



ensemble mean shows a tendency towards a drying with less precipitation in summer, especially in SWL2. However, more than half of the GCMs and RCMs actually show increasing precipitation and it is clear that the ensemble average is heavily influenced by a smaller number of models with relatively strong decreases. Furthermore, several of these models also show a strong warming indicating a feedback mechanism including reduced soil moisture. As for Scandinavia the spread becomes

larger at SWL2 compared to SWL1.5 and again we note that the RCM-simulated changes in temperature and precipitation mostly lie within the range of those as simulated by the underlying GCMs and by the larger CMIP5 ensemble. However, there are differences of which the most notable is that none of the RCMs gives a strong drying and warming in this region. This is the case even for the three RCMs downscaling HadGEM2-ESM. Clearly, the RCMs change the summertime climate change signal in this region in a significant way resulting in both a smaller signal and less spread than that seen in the GCMs.

Summary statistics for the ensembles including minimum, maximum, standard deviation and mean values for the regions in Fig. 1 are shown in Table 3. The numbers reveal that the three ensembles are different both for temperature and precipitation. Evidently, the smaller nine-member ensemble of GCMs that have been downscaled show less spread between minimum and maximum compared to the larger 31-member CMIP5 ensemble from where they are taken. However, we note

that the difference in spread as defined by one standard deviation is relatively small and the intervals always overlap. A systematic difference is that the ensemble mean temperature increases are lower in the nine-member GCM subset compared to the full CMIP5 ensemble by between 0.06 and 0.29°C for all eight regions in DJF or JJA. For SWL2 the same is found with corresponding differences in the range 0.14 to 0.36°C. These differences seem to be caused by  a number of GCMs with relatively strong response that has not been downscaled by any RCM (cf. Fig. 9). For precipitation we cannot find any

similar systematic differences. Rather, the subset is sometimes simulating wetter (or less dry) future conditions and sometimes the opposite.

Next we compare the RCM simulated climate change signal with that of the underlying nine-member GCMs. Again, we note that the ensembles differ. Here, we see that the RCMs tend to give smaller increases in temperature and larger increases in

precipitation (or less drying) than the GCMs. The differences in temperature ranges are most pronounced on the warmer side with substantially lower maximum warming in both summer and winter. The smaller spread between the ensemble members for the RCM simulations when it comes to temperature can also be seen in terms of lower standard deviation. That the RCMs are modifying the climate change results compared to the underlying GCMs, is also found in Sørland et al. (2017). Despite these changes we still note that in all seasons and all regions, the ranges given by the ensemble means plus/minus one

standard deviation overlap each other for both temperature and precipitation in all regions and for all seasons.

The results presented here indicate that: i) the RCMs changes the climate change signal compared to the GCMs they have been downscaling, ii) the RCM ensemble is within the range of the wider CMIP5 ensemble for seasonal mean temperature





and precipitation on the regional level, iii) a different sampling of the CMIP5 ensemble would lead to different results in the resulting RCM ensemble with implication on experimental design for impact studies.

**5 Conclusions**

The results show that simulated changes in temperatures indicate that Europe will warm in all seasons in the future and that these increases in temperature are highly significant and robust over the ensemble despite considerable natural variability in the climate. Consequently, already at the SWL1.5, we note increasing temperature in all European areas in a vast majority of the simulations. The simulated temperature changes in Europe are mostly larger than the global mean warming. This is most pronounced in northern and northeastern Europe in winter and in southernmost and northernmost Europe in summer where warming is strongest. In these areas future temperature changes w.r.t. 1971-2000 are larger than respectively +1.5°C and +2°C at SWL1.5 and SWL2 which corresponds to a warming of almost +2°C respectively +2.5°C compared to pre-industrial (1861-1890) conditions.

The results indicate that precipitation will increase in most of Europe on an annual mean basis although with larger uncertainty than in temperatures. The current findings support earlier findings of more pronounced increases in all of Europe in winter and increases only in the north in summer when large parts of southern Europe is simulated to get less precipitation. At SWL1.5 changes are still relatively small with a spread between simulations that encompass zero change. At SWL2 larger more significant and robust changes are seen both in winter and summer.

Robust patterns of changing wind speed are only found over parts of the Atlantic region where wind speed tend to decrease. Here we note that there is only little (if any) coherence between different simulations and it stands clear that future changes in wind speed are highly uncertain. Naturally, there is a strong impact of changes in the MSLP (i.e. large-scale circulation) on regional wind changes. We also find strong regional/local changes in some other areas most notably oceanic areas in the north including the Arctic Ocean and parts of the Baltic Sea. We speculate that the wind speed increases in these areas are related to decreases in sea ice extent with consequent changes in stability conditions in the planetary boundary layer.

Changes in MSLP not only influences wind speed but also modifies the climate change signal in temperature and precipitation. Examples of this include: i) changes in precipitation across the Scandinavian mountains with increases along the western side in connection to a stronger north-south MSLP gradient over the northern Atlantic in the northern part of the model domain and vice versa and ii) modifications of the warming signal with lower than average warming in southern Europe in simulations when the storm tracks are displaced towards the north.

We note that the RCMs can alter the results of the GCMs leading either to amplification or attenuation of the climate change signal in the underlying GCMs. For the EURO-CORDEX ensemble it is clear that the RCMs tend to produce less warming

and more precipitation (or less drying) compared to the underlying GCMs in many areas in both winter and summer. The temperature results indicate that the RCM ensemble reduces the spread compared to the underlying GCMs. Furthermore, the chosen subset of GCMs is giving a slightly weaker increase in temperature compared to that of the larger full CMIP5 ensemble. In particular, the subset has relatively fewer members showing strong warming in the region. Despite this we

conclude that the spread represented by the standard deviations in the ensembles do overlap for all regions and seasons for both near-surface temperature and precipitation.

## 6 Data availability

The main web page documenting data availability for EURO-CORDEX data can be found at http://euro-

cordex.net/060378/index.php.en. In general all CMIP5 and EURO-CORDEX simulations that have been analysed here are accessible via the international Earth System Grid Federation (ESGF). HadCRUT4 data analysed for global mean temperature changes was downloaded at https://www.metoffice.gov.uk/hadobs/hadcrut4/.

## 7 Competing interests

The authors declare that they have no conflict of interest.

## 8 Acknowledgements

Part of the work was done in the European Union Seventh Framework Programme in projects IMPACT2C project under grant agreement 282746 and HELIX under grant agreement 603864. Part of this work was done within the Swedish research program MSB-HazardsSupport. We acknowledge the World Climate Research Programme's Working Group on Coupled Modelling, which is responsible for CMIP, and we thank the climate modeling groups (listed in Table 2 of this paper) for

producing and making available their model output. For CMIP the U.S. Department of Energy's Program for Climate Model Diagnosis and Intercomparison provides coordinating support and led development of software infrastructure in partnership with the Global Organization for Earth System Science Portals.





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



**Table 1. Regional climate model simulations assessed in this report. GCMs are listed in more detail in Table 2.**

| No | Institute | RCM | GCM | RCM reference |
|----|-----------|-----|-----|---------------|
| 1 | SMHI | RCA4 | EC-EARTH-r12 | Kjellström et al. (2016) |
| 2 | | | HadGEM2-ES | |
| 3 | | | MPI-ESM-LR-r1 | |
| 4 | | | CNRM-CM5 | |
| 5 | | | IPSL-CM5A-MR | |
| 6 | BTU Cottbus | CCLM4-8-17 | EC-EARTH_r12 | Keuler et al. (2016) |
| 7 | | | CNRM-CM5 | |
| 8 | | | MPI-ESM-LR-r1 | |
| 9 | ETH | CCLM4-8-17 | HadGEM2-ES | Keuler et al. (2016) |
| 10 | HZG-GERICS | REMO2009 | MPI-ESM-LR-r1 | Jacob et al. (2012) |
| 11 | | | MPI-ESM-LR-r2 | |
| 12 | KNMI | RACMO2.2 | EC-EARTH-r1 | Meijgaard et al. (2012) |
| 13 | | | EC-EARTH-r12 | |
| 14 | | | HadGEM2-ES | |
| 15 | DMI | HIRHAM5 | EC-EARTH-r3 | Christensen et al. (1998) |
| 16 | | | NORESM1-M | |
| 17 | CNRM | ALADIN53 | CNRM-CM5 | Colin et al. (2010); Bador et al. (2017) |
| 18 | IPSL | WRF3.3.1 | IPSL-CM5A-MR | Skamarock et al. (2008) |



**Table 2. CMIP5 GCMs assessed here. Columns SWL1.5 and SWL2 show the central year in a 30-year period when GCMs reach the 1.5°C and 2°C warming levels (i.e. 2030 represents 2016-2045) under RCP8.5. GCMs are listed in order of when they reach SWL2. Only ensemble member r1 have been used unless otherwise noted in brackets after GCM name. GCMs in italics have been downscaled by RCMs (see Table 1). For more information see Taylor et al (2012) and http://cmip-pcmdi.llnl.gov/cmip5/.**

| No | Institute | GCM name | SWL 1.5 | SWL 2 |
|----|-----------|----------|---------|-------|
| 1 | Beijing Normal University | BNU-ESM | 2009 | 2023 |
| 2 | Canadian Centre for Climate Modelling and Analysis | CanESM2 | 2013 | 2026 |
| 3 | Institut Pierre-Simon Laplace | IPSL-CM5A-LR | 2011 | 2027 |
| 4 | Atmosphere and Ocean Research Institute (The University of Tokyo), National Institute for Environmental Studies, and Japan Agency for Marine-Earth Science and Technology | MIROC-ESM | 2020 | 2030 |
| 5 | | MIROC-ESM-CHEM | 2018 | 2030 |
| 6 | National Center for Atmospheric Research | CCSM4 | 2013 | 2030 |
| 7 | *Institut Pierre-Simon Laplace* | *IPSL-CM5A-MR* | *2016* | *2030* |
| 8 | *Max Planck Institute for Meteorology* | *MPI-ESM-LR (r2)* | *2016* | *2032* |
| 9 | NASA/GISS (Goddard Institute for Space Studies) | GISS-E2-H-CC | 2017 | 2035 |
| 10 | *EC-EARTH consortium* | *EC-EARTH (r1)* | *2017* | *2035* |
| 11 | Geophysical Fluid Dynamics Laboratory | GFDL-CM3 | 2023 | 2035 |
| 12 | *Max Planck Institute for Meteorology* | *MPI-ESM-LR (r2)* | *2018* | *2035* |
| 13 | *EC-EARTH consortium* | *EC-EARTH (r12)* | *2019* | *2035* |
| 14 | NASA/GISS (Goddard Institute for Space Studies) | GISS-E2-H | 2020 | 2036 |
| 15 | *EC-EARTH consortium* | *EC-EARTH (r3)* | *2020* | *2037* |
| 16 | Institut Pierre-Simon Laplace | IPSL-CM5B-LR | 2022 | 2037 |
| 17 | *Met Office Hadley Centre* | *HadGEM2-ES* | *2024* | *2037* |
| 18 | Max Planck Institute for Meteorology | MPI-ESM-MR | 2020 | 2038 |
| 19 | Met Office Hadley Centre | HadGEM2-CC | 2029 | 2041 |
| 20 | The First Institute of Oceanography, SOA | FIO-ESM | 2027 | 2042 |
| 21 | *Centre National de Recherches Météorologiques / Centre Européen de Recherche et Formation Avancée en Calcul Scientifique* | *CNRM-CM5* | *2029* | *2043* |
| 22 | Commonwealth Scientific and Industrial Research Organization/Queensland Climate Change Centre of Excellence | CSIRO-Mk3-6-0 | 2032 | 2044 |
| 23 | Met Office Hadley Centre | HadGEM2-AO | 2034 | 2046 |
| 24 | Norwegian Climate Centre | NorESM1-ME | 2032 | 2046 |
| 25 | NASA/GISS (Goddard Institute for Space Studies) | GISS-E2-R-CC | 2031 | 2048 |
| 26 | *Norwegian Climate Centre* | *NorESM1-M* | *2033* | *2048* |
| 27 | Atmosphere and Ocean Research Institute (The University of Tokyo), National Institute for Environmental Studies, and Japan Agency for Marine-Earth Science and Technology | MIROC5 | 2033 | 2048 |
| 28 | NOAA Geophysical Fluid Dynamics Laboratory | GFDL-ESM2M | 2034 | 2051 |
| 29 | Meteorological Research Institute | MRI-CGCM3 | 2040 | 2052 |
| 30 | Geophysical Fluid Dynamics Laboratory | GFDL-ESM2G | 2037 | 2054 |
| 31 | Russian Academy of Sciences, Institute of Numerical Mathematics | inmcm4 | 2043 | 2058 |



**Table 3. Summary statistics showing temperature and precipitation changes at SWL1.5 for the eight regions in Fig. 1. For each region there are three sets of data for each season and variable representing: the full CMIP5 ensemble (top), the nine-member GCM-ensemble downscaled by the RCMs (middle) and the eighteen-member RCM ensemble (lower). The numbers represents minimum (left), maximum (right) and mean plus/minus one standard deviation (middle).**

| Area | Near surface temperature (°C) | | Precipitation (%) | |
|------|------|------|------|------|
| | DJF | JJA | DJF | JJA |
| IP | 0.35 / 0.92±0.35 / 1.67 | 0.58 / 1.42±0.55 / 2.68 | -17 / -2.0±8.9 / 19 | -33 / -7.8±8.6 / 13 |
| | 0.40 / 0.87±0.40 / 1.67 | 0.74 / 1.16±0.47 / 1.99 | -3.6 / 5.6±6.8 / 19 | -21 / -7.3±8.6 / 9.4 |
| | 0.35 / 0.79±0.40 / 1.58 | 0.58 / 0.94±0.32 / 1.59 | -2.7 / 3.5±5.3 / 19 | -16 / -6.0±5.3 / 3.9 |
| MD | 0.37 / 1.02±0.44 / 1.95 | 0.65 / 1.54±0.59 / 2.75 | -19 / -3.0±6.8 / 10 | -30 / -6.3±9.8 / 15 |
| | 0.37 / 0.94±0.44 / 1.75 | 0.72 / 1.36±0.54 / 2.30 | -8.1 / 1.4±6.4 / 10 | -28 / -11±8.1 / -1.5 |
| | 0.27 / 0.92±0.43 / 1.77 | 0.66 / 1.16±0.36 / 1.85 | -7.7 / 1.0±5.6 / 12 | -18 / -1.7±9.0 / 15 |
| FR | 0.44 / 1.01±0.44 / 2.04 | 0.23 / 1.33±0.63 / 2.65 | -13 / 3.4±5.8 / 13 | -27 / -5.3±9.5 / 15 |
| | 0.46 / 0.84±0.41 / 1.63 | 0.44 / 1.05±0.59 / 2.13 | -9.1 / 3.6±6.8 / 11 | -12 / -5.1±7.4 / 12 |
| | 0.25 / 0.83±0.41 / 1.46 | 0.51 / 0.90±0.33 / 1.81 | -6.7 / 4.5±6.0 / 12 | -11 / -2.0±7.7 / 11 |
| AL | 0.28 / 1.29±0.64 / 2.91 | 0.50 / 1.65±0.74 / 3.43 | -11 / 3.6±7.6 / 16 | -29 / -1.3±9.2 / 26 |
| | 0.40 / 1.12±0.69 / 2.59 | 0.54 / 1.45±0.69 / 2.63 | -3.6 / 5.4±7.1 / 15 | -8.6 / -1.7±4.6 / 4.8 |
| | 0.24 / 1.07±0.53 / 2.00 | 0.73 / 1.15±0.30 / 1.86 | -6.8 / 5.2±5.5 / 12 | -13 / -0.5±6.6 / 10 |
| EA | -0.23 / 1.49±0.77 / 3.54 | 0.45 / 1.76±0.87 / 4.02 | -12 / 4.8±5.6 / 14 | -23 / 0.3±9.3 / 16 |
| | -0.23 / 1.21±0.77 / 2.30 | 0.51 / 1.53±0.85 / 3.26 | -12 / 4.4±7.3 / 11 | -16 / 0.2±9.5 / 16 |
| | -0.21 / 1.14±0.74 / 2.07 | 0.40 / 1.09±0.44 / 1.83 | -8.6 / 6.0±6.6 / 13 | -7.5 / 2.1±4.6 / 11 |
| BI | 0.09 / 0.81±0.36 / 1.75 | -0.15 / 0.97±0.52 / 1.97 | -2.0 / 4.7±4.6 / 17 | -16 / -1.1±7.0 / 16 |
| | 0.09 / 0.70±0.32 / 1.06 | 0.39 / 0.83±0.45 / 1.80 | -2.0 / 2.7±4.0 / 9.5 | -12 / 0.0±7.7 / 12 |
| | 0.11 / 0.71±0.28 / 1.02 | 0.35 / 0.83±0.34 / 1.60 | -4.8 / 3.5±4.4 / 11 | -5.4 / 1.0±4.7 / 7.4 |
| ME | 0.14 / 1.24±0.56 / 2.71 | 0.08 / 1.41±0.73 / 3.27 | -13 / 6.4±6.1 / 15 | -18 / 0.9±9.8 / 23 |
| | 0.14 / 0.98±0.54 / 1.85 | 0.35 / 1.14±0.70 / 2.59 | -13 / 3.3±7.8 / 12 | -6.8 / 2.6±8.9 / 21 |
| | 0.08 / 0.94±0.51 / 1.67 | 0.40 / 1.00±0.37 / 1.85 | -11 / 4.3±7.6 / 13 | -8.6 / 1.5±5.6 / 11 |
| SC | -0.06 / 1.67±0.71 / 3.10 | 0.16 / 1.45±0.66 / 2.83 | -4.0 / 5.4±4.8 / 16 | -4.7 / 4.2±5.5 / 13 |
| | 0.23 / 1.42±0.55 / 2.10 | 0.35 / 1.25±0.66 / 2.50 | -4.0 / 1.6±3.3 / 5.3 | -3.9 / 5.8±5.9 / 13 |
| | 0.22 / 1.53±0.44 / 2.00 | 0.52 / 1.25±0.51 / 2.20 | -5.4 / 3.2±3.3 / 6.8 | -1.1 / 5.3±3.1 / 10 |



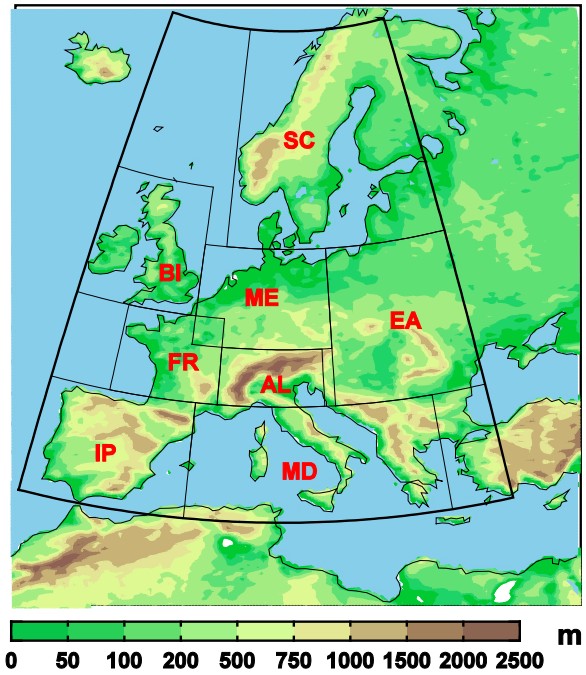

**Figure 1.** Map showing the eight subdomains (BI – the British Isles, IP – the Iberian Peninsula, FR – France, ME – Mid Europe, SC – Scandinavia, MD – the Mediterranean region, AL – the Alps, EA – Eastern Europe) and the larger European domain for which average climate change signals have been calculated. The colors represent the altitude of the surface in the models.





**Figure 2. Winter (DJF) mean sea level pressure in the reference period (uppermost left) and its change in seventeen RCM simulations in Table 1 (individual runs in upper right panel and all other rows and ensemble mean in second upper panel from the left) for the +2°C warming level (SWL2).**







**Figure 3.** Summer (JJA) mean sea level pressure in the reference period (uppermost left) and its change in seventeen RCM simulations in Table 1 (individual runs in upper right panel and all other rows and ensemble mean in second upper panel from the left) for the +2°C warming level (SWL2).





**Figure 4.**
Winter (DJF) 2m-temperature (top), precipitation (middle) and 10m-wind speed (lower) in the control period (left), its change at SWL1.5 (second column) and SWL2 (third column), and the difference between the change at SWL2 and SWL1.5 (rightmost column). Hatching in the climate change signal for precipitation and wind speed represents areas where at least 14 of the 18 ensemble members agree on the sign of change (for temperature this is always the case). Hatching in the rightmost plots indicates that changes at SWL2 are larger than those at SWL1.5 in at least 14 of the models.



**Figure 5.** Summer (JJA) 2m-temperature (top), precipitation (middle) and 10m-wind speed (lower) in the control period (left), its change at SWL1.5 (second column) and SWL2 (third column), and the difference between the change at SWL2 and SWL1.5 (rightmost column). Hatching in the climate change signal for precipitation and wind speed represents areas where at least 14 of the 18 ensemble members agree on the sign of change (for temperature this is always the case). Hatching in the rightmost plots indicates that changes at SWL2 are larger than those at SWL1.5 in at least 14 of the models.



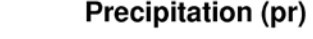

**Figure 6. Winter (DJF) precipitation in the reference period (uppermost left) and its change in eighteen RCM simulations in Table 1 (ensemble mean in the uppermost two left panels in the upper row all other panels are individual runs) for the +2°C warming level (SWL2).**





**Figure 7. Winter (DJF) 10m-wind speed in the reference period (uppermost left) and its change in eighteen RCM simulations in Table 1 (ensemble mean in the uppermost two left panels in the upper row all other panels are individual runs) for the +2°C warming level (SWL2).**





**Figure 8. Temperature and precipitation changes over Scandinavia (SC, Fig. 1) for winter (upper row) and summer (lower) mean conditions. The left plot shows SWL1.5 and the right SWL2. The error bars plotted inside the axis in the diagram illustrates the average and plus-minus one standard deviation from respectively: i) the CMIP5 ensemble (Table 2), ii) the nine-member GCM-ensemble that has been downscaled and iii) the eighteen-member RCM-ensemble (Table 1). Unfilled circles are CMIP5 GCMs listed in Table 2 that have not been downscaled. Filled circles represent GCMs that have been downscaled and these are connected by a line to the RCM(s) that have been used for downscaling. The horizontal line represents the global mean warming at resp. SWL1.5 and SWL2 relative to the control period (1971-2000).**





**Figure 9. Temperature and precipitation changes over Eastern Europe (EA, Fig. 1) for winter (upper row) and summer (lower) mean conditions. The left plot shows SWL1.5 and the right SWL2. The error bars plotted inside the axis in the diagram illustrates the average and plus-minus one standard deviation from respectively: i) the CMIP5 ensemble (Table 2), ii) the nine-member GCM-ensemble that has been downscaled and iii) the eighteen-member RCM-ensemble (Table 1). Unfilled circles are CMIP5 GCMs listed in Table 2 that have not been downscaled. Filled circles represent GCMs that have been downscaled and these are connected by a line to the RCM(s) that have been used for downscaling. The horizontal line represents the global mean warming at resp. SWL1.5 and SWL2 relative to the control period (1971-2000).**