# Peer review of "European climate change at global mean temperature increases of 1.5 and 2°C above pre-industrial conditions as simulated by the EURO-CORDEX regional climate models"

_Earth System Dynamics, 2017_

## Referee Comment (RC1) · Anonymous Referee #1 · 14 Dec 2017

This manuscript investigated European regional climate change at global mean temperature increased by 1.5 oC or 2 oC above pre-industrial conditions based on ERUO-CORDEX regional downscaling. Results showed that regional warming exceeds the global mean temperature in most parts of Europe while precipitation increased in the north of Europe and decreased in the south with larger uncertainty relative to those of temperature. The changes in temperature, precipitation and wind speed were shown modified by changes in mean sea level pressure indicating a strong relationship with the large-scale circulation and its internal variability on decade-long timescales.

[Figure]

It's an interesting topic but more deep analysis and discussion should be done. This manuscript adopted 31 CMIP5 modes just for calculating SWL1.5 and SWL2. The temperature, precipitation, and wind based on these CMIP5 modes might also be addressed to better show the differences with results based on RCMs. Additionally, in this manuscript, changes in temperature, precipitation and wind speed in Europe were attributed to changes in mean sea level pressure which was indicating a strong relationship with the large-scale circulation, but I think more discussion (such as humidity, wind profile, etc) is needed to support the conclusion. Finally, the overall quality of the manuscript should also be improved. Thus, careful and rigorous major revision is needed to bring the manuscript up to the standards for ESD.

List of specific (major and minor) comments:

Page 4, Line 24: Do you mean the other CMIP5 GCMs out of 31 selected models? Page 4, Line 28: It seems not suitable to say "RCMs change the climate change signal of the underlying GCMs" Page 4, Line 28: a large number of studies cited by Rockel and Woth ( 2017)? If not, please give more related citation. Page 5, Line 2, Line 6: Maybe "1.5 or 2.0 oC" is better. Page 5, Lines 8-10: The global warming between the pre-industrial and reference period based on observation (HadCRU4) is 0.41K. Does it better to calculate such global warming between these two periods for each CMIP5 model separately? Thus, each RCM could present the regional warming under the future temperature change above 1.5 or 2 oC subtracting the forcing GCM's warming between the two periods. Additionally, please change "0.41K" to "0.41 oC" to keep the unit consistent. Page 5, Line 20: There are too many subfigures in one figure. It's better to assign numbers to them and cite the subfigure in the main body. Same problems were found for other figures. Page 7, Line 9: The leftmost and rightmost colors of the label bar are too similar. Please revise the label bars of all related figures to make the spatial pattern clear to the readers. Page 7, Lines 11-12: Please give more detailed discussion such as horizontal wind. Page 8, Lines 1-11: When you discuss the connection between the precipitation and MSLP. Please give more discussion since

precipitation is high related to vertical and horizontal wind, humidity, etc. Page 8, Line 12: As a vector, wind direction is also important as well as wind speed. In section 3.4, why only wind speed discussed? Page 11, Line 3: Do you mean Table 2? Page 11, Line 18: Do you mean Table 2? Page 22, Table2: What's the meaning of the italic GCMs in Table 2. Page 23, Table 3: Please do not use "/" to separate the data since it's usually a sign of division. Page 24, Figure 1: Please present the latitude and longitude for the map. Same problems for other spatial plots. Pages 25-26, Figures 2-3: "seventeen RCM simulations", it seems 18 RCMs used in this study. Please give significant test of the diferences if possible. Additionally, please explain in the main body why the subfigure of WRF is blank.

Technical corrections:

Page 3, Line 28: works Page 3, Line 31: ; should be , Page 5, Line 16: two "."

---

## Referee Comment (RC2) · Anonymous Referee #2 · 2 Jan 2018

The authors investigated the climate changes over European region at 1.5C and 2C warming under RCP8.5 scenario mainly based on the EURO-CORDEX regional climate models simulations. The possible changes in seasonal mean temperature, precipitation and surface wind at different warming target were described and compared to those from the corresponding driving global climate models. This work is timely, and may be useful for the coming IPCC special report. My comments are as follows. (1) After I read through the whole manuscript, it is not clear to me whether it is beneficial for Europe to control the warming target to 1.5°C rather than 2°C. In the manuscript,

the authors always talked about the possible climate changes at different warming target, but how about the corresponding differences between 1.5C and 2C warming? The authors showed the differences in Fig.4 and 5, but they didn't discuss them. For example, are there any differences statistically significant (the model agreements do not mean the statistical significance)? (2) In Fig.8 and 9, the authors compared the results under 1.5C and 2C warming derived from regional climate models and global climate models. Is it possible to show us the differences between the 1.5C and 2C warming from different models? (3) The authors mainly focused on the seasonal mean changes, and the changes in precipitation and wind are quite uncertain. How about the changes in extreme events (precipitation and wind)? The manuscript could be more interesting if the authors included some analysis on the changes in extreme climate events, and discuss whether the 0.5C less warming could reduce significantly the extreme climate events in European region, based on the regional climate models with high resolution. Minor comments: (4) Figure 2 and 3: why is there no result from WRF (IPSL)? (5) L15-20: "attenuation of amplification of" should be "attenuation or amplification of" (6) In the abstract, you should mention that this study focus on climate changes at different warming target under RCP8.5 scenario. The scenario information is very important, since the conclusions may be scenario-dependent.

---

## Author Comment (AC1) · 21 Feb 2018

**Revision of the paper and response to reviewer comments**

We would like to thank the two reviewers for their valuable comments on the manuscript. In the text below we have indicated how we have addressed the comments (the *reviewer comments are included in italics* and our response is given in red and revised specific sentences are given in blue). The main changes that we have introduced in the paper are:

- The introduction of a measure of robustness of the climate change signal by illustrating the signal-to-noise ratio defined by dividing the ensemble average signal with the ensemble spread (measured by one standard deviation). This is done only in areas where the climate change signal is consistent as defined by at least 80% of the simulations showing the same sign of the change. This is described in the new section 2.3 in the paper.
- Addition of a table illustrating for how large fraction of Europe the models agree on the sign of the climate change signal and show a significant signal-to-noise ratio.
- Inclusion of some more results from the underlying GCMs to the discussion.

**Reply to comments made by Reviewer 1**

*This manuscript investigated European regional climate change at global mean temperature increased by 1.5 oC or 2 oC above pre-industrial conditions based on ERUOCORDEX regional downscaling. Results showed that regional warming exceeds the global mean temperature in most parts of Europe while precipitation increased in the north of Europe and decreased in the south with larger uncertainty relative to those of temperature. The changes in temperature, precipitation and wind speed were shown modified by changes in mean sea level pressure indicating a strong relationship with the large-scale circulation and its internal variability on decade-long timescales.*

*It's an interesting topic but more deep analysis and discussion should be done. This manuscript adopted 31 CMIP5 modes just for calculating SWL1.5 and SWL2. The temperature, precipitation, and wind based on these CMIP5 modes might also be addressed to better show the differences with results based on RCMs.*

We acknowledge this comment about adding more results from the GCMs as it is interesting to i) show the climate change signal in a wider ensemble and ii) discuss if and to what extent the RCMs alter the climate change signal in the GCMs.

Even if this is not the main focus of this paper that concentrates on the RCM results some aspects of climate change (seasonal mean changes in temperature and precipitation) in the GCMs were shown already in the original version (Figures 8 and 9) where the RCM results are put in a wider context and where it can clearly be seen that the RCMs alter the climate change signal of the GCMs.

In addition to what was present in the originally submitted manuscript we have produced a set of additional figures showing ensemble mean changes and robustness estimates in the same way as Figure 4 and 5 but instead based on the underlying GCMs. These figures are added in the supplementary material and allow the reader to see the geographical patterns of the GCM signal and to what extent the agreement within the RCM and GCM ensembles are similar.

*Additionally, in this manuscript, changes in temperature, precipitation and wind speed in Europe were attributed to changes in mean sea level pressure which was indicating a strong relationship with the large-scale circulation, but I think more discussion (such as humidity, wind profile, etc) is needed to support the conclusion.*

We don't agree that we make a strong conclusion about attribution of climate change to changes in MSLP. Rather we state that MSLP changes modify the large-scale climate change signal seen in other variables. The fact that such changes in large-scale circulation as manifested by changes in MSLP have a strong impact on other variables is not a new finding of this study but well-established knowledge and we have chosen not to expand the paper by discussing it in more detail here. We have reformulated the sentence where we motivate why we choose to show the MSLP changes and also added some references to it so that it now reads:

First, however, we show how changes in mean sea level pressure (MSLP) differ between the individual ensemble members as these changes are known to have strong impacts on changes in the other variables (e.g. Van Ulden and van Oldenborgh, 2006; Kjellström et al., 2011; Aalbers et al., 2017).

*Finally, the overall quality of the manuscript should also be improved. Thus, careful and rigorous major revision is needed to bring the manuscript up to the standards for ESD.*

We have carefully revised the manuscript taken into account all comments by both reviewers.

*List of specific (major and minor) comments:*

*Page 4, Line 24: Do you mean the other CMIP5 GCMs out of 31 selected models?*

We have added numbers to explicitly state how many GCMs we are referring to.

"In addition to the nine GCM simulations listed in Table 1, also the first ensemble members of the other 22 CMIP5 GCMs are assessed"

*Page 4, Line 28: It seems not suitable to say "RCMs change the climate change signal of the underlying GCMs"*

The sentence has been modified to

"RCMs modify the climate change signal compared to the underlying GCMs"

*Page 4, Line 28: a large number of studies cited by Rockel and Woth ( 2017)? If not, please give more related citation.*

We have added a few studies here. The paper by Rockel and Woth is highly relevant in this context as this is the paper where these areas were first defined.

*Page 5, Line 2, Line 6:*
*Maybe "1.5 or 2.0 oC" is better.*

Indeed, corrected.

*Page 5, Lines 8-10: The global warming between the pre-industrial and reference period based on observation (HadCRU4) is 0.41K. Does it better to calculate such global warming between these two periods for each CMIP5 model separately? Thus, each RCM could present the regional warming under the future temperature change above 1.5 or 2 oC subtracting the forcing GCM's warming between the two periods.*

This is an interesting consideration and it is by no means clear how to best define these periods. As explained in the text each individual GCM is screened for when 1.5 and 2 C above 1861-1890 is first found. This time period is then used for analysis also for the individual RCMs. The last sentence of this paragraph simply states what the observed temperature increase is between "preindustrial (1861-1890)" and "reference (1971-2000)". An alternative would be to screen all individual GCMs for when they reach first 0.41 C and then 1.5/2 C above preindustrial conditions as we interpret the review comment. This would leave us with comparing a set of simulations with different time periods both in the beginning (e.g. 1960-1989, 1975-2004, etc.) and in the end (see Table 2). It is not evident that such an approach would lead to any clear benefit and it could be an area of further investigation. Here, most RCM simulations do not start before 1970 which means that we would simply not have data for any earlier reference period. We have therefore chosen not to make any changes apart from adding the following sentence explaining this to the method section.

"This choice is made as i) the starting point (1971) is the first possible as not all RCMs have data for earlier years and ii) the end point (2000) is before the first year in any of the 30-year SWL1.5 time periods downscaled here (the IPSL model, number 7 in Table 2)."

*Additionally, please change "0.41K" to "0.41 oC" to keep the unit consistent.*

Done

*Page 5, Line 20: There are too many subfigures in one figure. It's better to assign numbers to them and cite the subfigure in the main body. Same problems were found for other figures.*

We agree that there are many subfigures in one figure. However, we do not agree that it is too many. In fact it is the purpose of these figures to show the different flavors of change signals in the different simulations. We have carefully worked with the design of these figures in order to place the different subfigures in a logical order so that the influence of choosing different GCMs or RCMs can easily be seen. This ordering of subplots is kept throughout the manuscript so that all figures showing individual ensemble members do so at the same subplot so that figures are easily comparable. Furthermore, all subplots are associated with a label stating which GCM/RCM-pair that is shown so that the reader can concentrate on looking at the figure instead of having to look back and forth between the subplot labels and the Figure caption.

*Page 7, Line 9: The leftmost and rightmost colors of the label bar are too similar. Please revise the label bars of all related figures to make the spatial pattern clear to the readers.*

We don't understand this comment. If it relates to the leftmost and rightmost colors in the central two subplots for each row these goes from dark bluish to dark red with an additional greenish color for numbers outside of the outermost numbers (e.g. -4 and +4 C) and there cannot be any question as to whether we are on the negative or positive side. Deliberately, the colours are a bit bleaker close to zero to indicate smaller change signal. If the comment relates to differences between the two leftmost panels and the rightmost one then we can only say that these panels are associated with completely different color scales that are given beneath the panels.

*Page 7, Lines 11-12: Please give more detailed discussion such as horizontal wind.*

We have reformulated the sentence slightly so that it now starts with MSLP and its direct consequences for the horizontal wind before going into the consequences for temperature.

Recalling the changes in MSLP (Fig. 2) with on average weaker southwesterlies over large parts of the North Atlantic we interpret this modest warming as a consequence of the changing large-scale circulation bringing less mild Atlantic air in over Europe.

*Page 8, Lines 1-11: When you discuss the connection between the precipitation and MSLP. Please give more discussion since precipitation is high related to vertical and horizontal wind, humidity, etc.*

We have modified the text slightly so that we now explicitly mention orographic amplification of the precipitation changes.

*Page 8, Line12: As a vector, wind direction is also important as well as wind speed. In section 3.4, why only wind speed discussed?*

We acknowledge that changes in wind direction can also be important. As we have already included the section on MSLP differences from where some inferences about changes in large-scale circulation including changes in wind direction can be made we have chosen to limit this section to showing results for wind speed. A natural thing to include when discussing wind direction is arrows illustrating the wind vectors (and/or changes in them) in a figure. For clarity reasons, as there are indeed many subplots in the figures (as remarked by the reviewer) we have chosen to refrain from this and concentrate on wind speed.

*Page 11, Line 3: Do you mean Table 2?*

Yes, corrected.

*Page 11, Line 18: Do you mean Table 2?*

Yes, corrected.

*Page 22, Table2: What's the meaning of the italic GCMs in Table 2.*

As stated in the Table caption "GCMs in italics have been downscaled by RCMs".

*Page 23, Table 3: Please do not use "/" to separate the data since it's usually a sign of division.*

We have removed the "/" and separated minimum, mean plus/minus one standard deviation, maximum in three different columns instead.

*Page 24, Figure 1: Please present the latitude and longitude for the map. Same problems for other spatial plots.*

We have included latitude and longitudes in Figure 1. We have, however, chosen not to include them in all subplots in the rest of the figures of the papers to keep the figures more easily readable. We are aware of the fact that there are many subplots as the reviewer points out and including more information in them is not helpful in this respect.

*Pages 25-26, Figures 2-3: "seventeen RCM simulations", it seems 18 RCMs used in this study.*

Figures 2-3 are based on 17 members as MSLP is not available for one of the models. This has now been explained in the main text in section 2 describing the climate model data.

*Please give significant test of the diferences if possible.*

We have added a paragraph addressing robustness and significance of the results (2.3). This is now also illustrated in the figures showing ensemble mean changes.

*Additionally, please explain in the main body why the subfigure of WRF is blank.*

We have added an explanation as to why we have not included MSLP data from WRF (as data is lacking). The panels are kept there in order to keep the figures organized in the same way as the other figures.

*Technical corrections:*

*Page 3, Line 28: works*

Corrected

*Page 3, Line 31: ; should be ,*

Corrected

*Page 5, Line 16: two "."*

Corrected

**Reply to comments made by Reviewer 2**

*The authors investigated the climate changes over European region at 1.5C and 2C warming under RCP8.5 scenario mainly based on the EURO-CORDEX regional climate models simulations. The possible changes in seasonal mean temperature, precipitation and surface wind at different warming target were described and compared to those from the corresponding driving global climate models. This work is timely, and may be useful for the coming IPCC special report. My comments are as follows.*

*(1) After I read through the whole manuscript, it is not clear to me whether it is beneficial for Europe to control the warming target to 1.5_C rather than 2_C. In the manuscript, the authors always talked about the possible climate changes at different warming target, but how about the corresponding differences between 1.5C and 2C warming? The authors showed the differences in Fig.4 and 5, but they didn't discuss them. For example, are there any differences statistically significant (the model agreements do not mean the statistical significance)?*

We have expanded the discussion about differences at different warming levels and to what extent a significant climate change signal emerges at different SWLs. For this we have used agreement in sign of climate change for the three variables indicating whether the signal is consistent or not. In the revision we have also introduced a measure of spread among those models agreeing upon change in sign so that the signal-to-noise ratio defined by the ensemble average climate change signal divided with the standard deviation among the ensemble members. When this ratio is larger than one, we say that we have a robust change. This is not a test of statistical significance but, in our opinion, a good way of showing ensemble agreement. We have added a new table showing for how large fraction of all land areas consistent and robust changes are seen at the different warming levels. In the revised version we also give supplementary material and we have added a table corresponding to Table 4 (Table 3 in the original submission) but now with the numbers for SWL2 for quantitative comparison between the two SWLs.

*(2) In Fig.8 and 9, the authors compared the results under 1.5C and 2C warming derived from regional climate models and global climate models. Is it possible to show us the differences between the 1.5C and 2C warming from different models?*

We think that adding also such difference plots would lead to too many figures and have therefore chosen not to show the individual difference plots but instead focused on the ensemble average and whether differences in it are robust or not. To further illustrate the ensemble mean characteristics we have also added another table in the Supplementary material showing the regional changes at SWL2 that can be compared to those for SWL1.5 given in Table 4.

*(3) The authors mainly focused on the seasonal mean changes, and the changes in precipitation and wind are quite uncertain. How about the changes in extreme events (precipitation and wind)? The manuscript could be more interesting if the authors included some analysis on the changes in extreme climate events, and discuss whether the 0.5C less warming could reduce significantly the extreme climate events in European region, based on the regional climate models with high resolution.*

We acknowledge that this would indeed be interesting. However, we also note that the number of various aspects of extreme conditions and changes in them are numerous and it is outside of the scope of this paper to show all these results. Instead it will be the topic of another study.

*Minor comments:*

*(4) Figure 2 and 3: why is there no result from WRF (IPSL)?*

Data is missing for MSLP for this model. This is now explained in the text.

*(5) L15-20: "attenuation of amplification of" should be "attenuation or amplification of"*

Corrected.

*(6) In the abstract, you should mention that this study focus on climate changes at different warming target under RCP8.5 scenario. The scenario information is very important, since the conclusions may be scenario-dependent.*

The information about forcing scenario has been added so that it is now clear that we only look at RCP8.5 simulations.